# Towards Understanding the Involvement of H^+^-ATPase in Programmed Cell Death of *Psammosilene tunicoides* after Oxalic Acid Application

**DOI:** 10.3390/molecules26226957

**Published:** 2021-11-18

**Authors:** Xinyu Jiang, Mohammad Aqa Mohammadi, Yuan Qin, Zongshen Zhang

**Affiliations:** 1Laboratory of Pharmaceutical Plant Cell Culture Research, School of Biological Engineering, Dalian Polytechnic University, Dalian 116034, China; jiangxinyu255@163.com; 2State Key Laboratory of Ecological Pest Control for Fujian and Taiwan Crops, Key Laboratory of Genetics, Breeding and Multiple Utilization of Crops, Ministry of Education, Fujian Provincial Key Laboratory of Haixia Applied Plant Systems Biology, College of Horticulture, Fujian Agriculture and Forestry University, Fuzhou 350002, China; 1161906001@m.fafu.edu.cn; 3Guangxi Key Laboratory of Sugarcane Biology, State Key Laboratory for Conservation and Utilization of Subtropical Agro-Bioresources, College of Agriculture, Guangxi University, Nanning 530004, China; 4Dalian Yike Biotechnology Co., LTD, Dalian 116016, China

**Keywords:** *Psammosilene tunicoides*, oxalic acid, H^+^-ATPase, programmed cell death

## Abstract

*Psammosilene tunicoides* is a unique perennial medicinal plant species native to the Southwestern regions of China. Its wild population is rare and endangered due to over-excessive collection and extended growth (4–5 years). This research shows that H^+^-ATPase activity was a key factor for oxalate-inducing programmed cell death (PCD) of *P. tunicoides* suspension cells. Oxalic acid (OA) is an effective abiotic elicitor that enhances a plant cell’s resistance to environmental stress. However, the role of OA in this process remains to be mechanistically unveiled. The present study evaluated the role of OA-induced cell death using an inverted fluorescence microscope after staining with Evans blue, FDA, PI, and Rd123. OA-stimulated changes in K^+^ and Ca^2+^ trans-membrane flows using a patch-clamp method, together with OA modulation of H^+^-ATPase activity, were further examined. OA treatment increased cell death rate in a dosage-and duration-dependent manner. OA significantly decreased the mitochondria activity and damaged its electron transport chain. The OA treatment also decreased intracellular pH, while the FC increased the pH value. Simultaneously, NH_4_Cl caused intracellular acidification. The OA treatment independently resulted in 90% and the FC led to 25% cell death rates. Consistently, the combined treatments caused a 31% cell death rate. Furthermore, treatment with EGTA caused a similar change in intracellular pH value to the La^3+^ and OA application. Combined results suggest that OA-caused cell death could be attributed to intracellular acidification and the involvement of OA in the influx of extracellular Ca^2+^, thereby leading to membrane depolarization. Here we explore the resistance mechanism of *P*. *tunicoides* cells against various stresses endowed by OA treatment.

## 1. Introduction

Oxalic acid (OA) is a simple dicarboxylic acid that widely exists in biological systems and plays functional role in plants. OA’s chemical nature as a potent metal chelator has received more attention for its physiological functions in metabolism and signalling pathways in plant cells [1]. Moreover, OA has been recorded as an effective elicitor, improving plants’ resistance against adverse effects of phytopathogens. For instance, OA induced systemic resistance of tomatoes against *Botrytis cinerea* and *Sclerotinia sclerotiorum* (*S. sclerotiorum*) in sunflower [2,3]. In addition, after plants are challenged to biotic stress, several oxidative and hydrolases activities in plant tissues occur [4,5,6], for example, chitinase, phenylalaninelyase, peroxidase, catalase, and β-1,3-glucanase [7,8,9]. Previous studies have shown that the exogenous application of OA on fungal culture filtrates containing secreted OA mimicked fungal disease prevalence in rice [10]. The secretion of OA was essential to affect pathogenicity in plants by *S. sclerotiorum* fungal pathogen infection [11]. Multiple observations further demonstrated that *S. sclerotiorum* mutants, deficient in OA synthesizing, were nonpathogenic [12]. The expression of genes encoding OA in wheat could enhance the host resistance against fungal invasion [13]. OA is one of the earliest and most universal plant defense responses related to oxidative response in the plant cell [12,14]. OA appears to function during plant-microbe interaction by triggering the pathways responsible for programmed cell death (PCD) in plants and may act as a signalling molecule [15]. The transduction of signals leading to the death of Arabidopsis cells in response to OA treatment was associated with the activity of the anion channel [16]. This death displayed characteristic hallmarks of PCD, such as cell shrinkage, de novo protein synthesis, cleavage of nuclear DNA, activation of anion channel-dependent, and gene expression [16].

OA could play critical role in regulating cellular Ca^2+^ concentration during physiological or pathological processes. Calcium oxalate, commonly representing more than 10% whole plant Ca, acts as a physiological, osmotic inactive product and often exists in large amounts in soybean plants [17]. At the same time, Ca^2+^ has been documented as a critical second messenger by which cells perceive and transmit extra- or intracellular stimulation. The previous study showed that the application of exogenous OA could modulate the distribution of Ca^2+^ in compartments of mesophyll cells and enhance plant resistance to heat and cold stresses [18,19]. Moreover, it has been observed that the dynamic changes in Ca^2+^ spatial and temporary distribution might correlate closely with its distinct roles played during PCD of plants or other plant physiology processes [20]. Mazen et al. [21] reported that Ca-oxalate in plant cells increased extracellular Ca^2+^ and not excess OA [21]. Further studies have demonstrated that calcium channels were involved in calcium oxalate crystal formation in specialized cells of porang corms at harvest time [22]. The above studies suggested the role of OA in plant physiological responses through the regulation of cellular Ca^2+^ concentration by formation of Ca-OA.

The electrochemical H^+^ gradient across the plasma membrane generated by the H^+^-ATPase is an essential feature of all plant cells. Both hormonal signals such as auxin and environmental factors can affect cell growth by inducing the cell to alter its wall pH through changes in the activity of H^+^-ATPases located in the plasma membrane [23]. Functions of transient alteration in [H^+^]_i_ or [H^+^]_o_ in the early response of plant cells to environmental stimuli, such as turgor, gravity, pathogen attack and chemicals exposure, have been well explored [24]. In addition, many physiological events of plant cells, such as nutrient transport across the plasma membrane, cell elongation, and organ development, are highly dependent on the ability of individual cells to control pH both in cytosol and apoplast [25]. Similar findings on function of intracellular pH alteration in the processes of animal cell growth, development, and survival have been presented from many aspects [26,27]. Generally, the modulation of intracellular pH (pHi) or extracellular pH (pH_o_) in plant or animal cells could lead to depolarization or hyperpolarization in the plasma membrane [28]. They were subsequently followed by triggering or inhibiting a series of physiological events at the plasma membrane, such as control of ion channels activities, signalling and nutrient uptake, and cell growth [29].

From the above studies, it is conceivable that OA is correlated to ion channels located at the plasma membrane in triggering responses of plant cells to various environmental perturbations. However, detailed experimental evidence has been insufficient. In this study, we confirmed that OA induced PCD in *P**. tunicoides* cells. The cytoplasmic pH (pHcyt) oscillations are essential in the process independent of the extracellular Ca^2+^. The study also suggested that OA may influence inward K^+^ channels and Ca^2+^ channels by mediating the activation of H^+^-ATPase.

## 2. Results

### 2.1. Effects of OA on Suspension Cell Viability of P. tunicoides

Previous studies have reported that differential expression of genes contributing to PCD triggered by exogenous OA in tomatoes [30]. The current study analyzed the effects of OA on suspension-cultured cells of *P. tunicoides* using live/dead staining methods with Evans blue and FDA (Figure 1). The percentage of cell death was measured every hour between 0–8 hours (h). It was found that 1 mM OA treatment had significant influence on plant cells death. The death rate of 90% ± 10 was observed after the 8 h of OA treatment. Compared with 100 µM OA treatment, the rate of cell death was about 54% ± 4 within 8 h, as shown in (Figure 1). Identical results were recorded even after 24 h of treatment. However, with decline of OA concentration to 10 µM, during the entire course of 8 h of treatment, no visible alteration in cell death rate could be observed. The evidence related to PCD, such as vacuole shrinkage, plasma membrane invagination and the formation of cysts, was clear and in agreement with published results [31,32]. These dose-dependent data supported published hypothesis that OA may influence the viability of cells to a measurable extent, and in some cases, even lead to PCD in *Panax ginseng* cells [33].

### 2.2. Effects of OA on Respiratory Electron Transporter Chains

It is common knowledge that mitochondria is the power-generating organelles of a cell. The respiratory electron transport chains provide the driving forces for metabolism and generate redox signals, regulating every aspect of plant biology by controlling enzyme gene expression [34,35]. Kinetic data indicate that mitochondrion undergoes significant changes in membrane integrity before classical signs of apoptosis manifest. These changes concern both inner and outer mitochondrial membranes, disrupting inner transmembrane potential and releasing inter-membrane proteins through the outer membrane [36]. This study examined the mitochondrial membrane potential (∆Ψ_m_) after treatment with 1 mM OA using the mitochondrial marker, rhodamine (Rd) 123. The decrease in green fluorescence intensity was observed in OA-treated cells after 15 min, becoming even more evident until 75 min, due to the collapse of ∆Ψ_m_ and loss of mitochondrial membrane integrity, as seen in (Figure 2). The diffuse, high level of cytoplasmic fluorescence was likely due to the loss of mitochondrial membrane integrity, and an inability to specifically accumulate Rd123 occurred in mitochondria under these conditions. The ratio of fluorescence intensities began to increase significantly in the cytoplasm. However, it could not reach the nadir as in its beginning. The images revealed that the mitochondrial self-repairing mechanism might work in the process, but the mitochondrial membrane integrity had already been lost and could not resume completely.

### 2.3. Effects of OA on Nuclear Membrane Integrity

In the early stage, cell membrane permeability is an essential index for distinguishing apoptosis-like PCD from necrosis. However, it will gradually further become leaky with the time-lapse of cells undergoing programmed death. Propidium iodide (PI), which only stains the nucleus of a late stage of PCD but is capable of degrading the nucleus of necrotic cells quickly, was employed and added to the media containing suspension cells treated by OA. As indicated in (Figure 3), the fluorescence intensity of the PI-staining nucleus began to appear only in the cells after 1 hour of OA stress and then increased and dispersed around the nucleus gradually (Figure 3B–E). Here, the staining of the nucleus represented the PI diffusion across the plasma membrane. However, its physical rupture did not occur after the loss of membrane integrity (Figure 3F). The result of PI staining demonstrated again the hallmark of PCD caused by OA, just as indicated by Rd123 staining of mitochondria.

### 2.4. Effects of OA on Cytoplasmic pH of P. tunicoides

To enhance normal cell function, pH_cyt_ oscillations are usually maintained within a narrow range. Several cellular processes, such as cytoskeletal organization, vesicle fusion, and enzyme activities, are sensitive to pH and might be regulated by differences in pH_cyt_ [37]. Cytosol acidification and the corresponding medium alkalinization are early events occurring in cells [38,39]. The pH values in rape oilseed decreased rapidly and were markedly lower than 5.63 measured before OA treatment [13]. Therefore, to characterize OA-induced events within plant cells, the question of whether the pH_cyt_ oscillations could be stimulated by OA or not was further sophisticated. A fluorescent indicator of pH_cyt_, BCECF-AM, which can release BCECF within the cells after hydrolysis by intracellular esterase, was used to detect the changes in pH_cyt_. The fluorescent intensity and emission from BCECF accumulated in the cytoplasm may change in a pH-dependent manner, and hence, pH_cyt_ can be mapped by analyzing fluorescence ratio imaging. OA-induced alterations in the fluorescence intensity and pH_cyt_ are shown in Figure 4. Once OA was added to the cell suspension (final concentration 1 mM and 100 μM), the fluorescence intensity began to decrease rapidly (from 0.132 ± 0.0086 to 0.07 ± 0.004 for 1 mM, and 0.131 ± 0.0062 to 0.097 ± 0.0058 for 100 μM, respectively) within 30 min, which meant pH_cyt_ began to drop drastically over a short time.

The results indicated that the treatment of OA led to acidification within the cytoplasm, which was dependent on its concentration. Furthermore, it was clear that 1 mM OA that influenced cytoplasmic acidification was more significant than 100 μM OA did. At the same time, pHcyt had no significant difference in the presence of 10 µM OA.

### 2.5. Effects of NH_4_Cl and Fusicoccinon on Cytoplasmic pH and Cell Viability of P. tunicoides

Previous studies have demonstrated that intracellular alkalization is probably associated with PCD or abscission of plant cells [40,41]. Two experimental models were established to explore further the physiological roles of cytoplasmic acidification in the progress of cell death following OA treatment. The first could decrease pHcyt, causing the temporary intracellular acidification without altering extracellular pH. The second could increase the pHcyt to relieve intracellular acidification. (Figure 5) illustrates pHcyt changes in 5 min application followed by removal of NH_4_Cl, compared with continuous application of NH_4_Cl. The present dose-dependent mode detected in the pHcyt was similar to that of the OA-induced cell death rate. After removal of NH_4_Cl, cytoplasmic acidification was indicated by the decrease of fluorescence intensity. Compared with the standard condition, the cell death rate was much higher under the cytoplasmic acidification treatment (data not shown). Fusicoccin (FC), a strong activator of the H^+^-ATPase, enhanced H^+^ export from the intracellular. FC may cause cytoplasmic alkalinization, and could weaken the pH_cyt_ alteration by OA as seen in (Figure 5A). Moreover, the addition of FC reduced the percentage of dead cells from 90.5 ± 5.7% to 31.61 ± 9.35% after being treated by OA. The results showed that cytoplasmic acidification was an inevitable precondition to the PCD induced by OA.

### 2.6. Effects of EGTA and La^3+^ on Cytoplasmic pH of OA-Treated Cells

Elicitor-induced Ca^2+^ spiking is one of the earliest events that act as a master messenger for almost all downstream response reactions. Medium alkalinization is thought to result from elicitor-induced depolarization of the plasma membrane and is associated with Ca^2+^ influx/Cl^−^efflux [39]. In *Chara coralline*, a lowered pH_cyt_ increased cytosolic free Ca^2+^ ([Ca^2+^]cyt) affinity, activating Cl^−^ efflux. OA could chelate Ca^2+^ to form Ca oxalate crystals. The formation is regarded as a highly controlled cellular process rather than a simple precipitation phenomenon. Specialized mechanisms must be present in crystal idioblasts to deal with the enormous fluxes of Ca^2+^ [42]. In order to investigate the role of extracellular Ca^2+^ in the process of acidification induced by OA, the cells were treated with ethylene glycol tetraacetic acid (EGTA) (a Ca^2+^ chelator) and La^3+^ (a Ca^2+^ channel blocker) before the addition of OA, respectively. From the results of fluorescent intensity, both EGTA and La^3+^ have a minor effect on the decrease of pH_cyt_ (Figure 6), which suggested extracellular Ca^2+^ may neither participate in the modification of the pH_cyt_ nor position downstream of acidification induced by OA.

### 2.7. The Changes Induced by OA in K^+^ and Ca^2+^ Inward

Exogenous molecules can modulate transporters located at the plasma membrane, and these elicitor-induced ion fluxes are immediate responses of plant cells. The cytoplasmic acidification and related medium alkalinization are thought to be due to elicitor-induced depolarization of the plasma membrane and subsequent K^+^/H^+^ exchanger, with Ca^2+^ influx, which is generally observed in the earliest responses of plant cells to avirulent pathogens [39]. In addition, the induction of this altered biochemical balance depends on an H^+^-pumping ATPase activity, and the stimulation of its activity and growth by FC and the inhibition by vanadate also support the idea that plasma membrane H^+^-ATPases play a role in the maintenance of pH_cyt_ [43,44]. The experiment used the electrophysiological approach to determine the position of OA in the alterations of K^+^, Ca^2+^ channel and H^+^-ATPases. The K^+^ channel and Ca^2+^ channel holding potential on *P. unicoides* cells are −50 mV and +20 mV, respectively, similar to Arabidopsis results in patch-clamp experiments. High dose OA, which caused evident PCD, induced a decrease of inward K^+^ current from 350 pA to 150 pA and increased inward Ca^2+^ wind from 50 pA to 300 pA after 10 min. To investigate the relationship between OA and H^+^-ATPase, the effects of H^+^-ATPases activator FC on the channel currents were also tested. The addition of FC, which effectively weakens the altered current process, caused inward K^+^ current to rise to 280 pA and inward Ca^2+^ current to fall to 150 pA, respectively. However, both channels’ current was altered significantly compared with that of the normal condition. These results suggested that the OA may regulate the inward K^+^ channel and inward Ca^2+^ channel by mediating H^+^-ATPases activity.

## 3. Discussion

Previous results demonstrated the accumulation of OA, which is essential for the pathogenicity of fungi. OA can acidify the infected plant tissues to activate many fungal enzymes and protein kinase of host plant cells at low pH and degrade the plant cell wall via acidity or chelation of the cell wall Ca^2+^ [37,38,39]. The research revealed that OA maintained its toxicity even the pH decline by OA treatment, and suggested that acidification was not the only mode of OA action bringing about deleterious effects during PCD in *A. thaliana* [16]. Likewise, we demonstrated that OA, even when its pH was adjusted equally to the suspension cells media, could induce PCD in *P. tunicoides*. These results suggested that OA itself functioned as an inducer of PCD; its acidity may accelerate the PCD process.

Recently, plant mitochondria as cellular stress sensors and central organelles in PCD have attracted increasing interest [45]. The outer organelle membrane disrupted and released proteins, such as cytochrome C (cyt*_c_*), into the cytosol, triggering caspase activation or performing other functions relevant to PCD activation of catabolic proteases and nucleases [46,47]. These changed cytosol circumstances and activated proteases and nucleases may influence the energy metabolism, disrupt the nuclear membrane, then break down the inside nuclear DNA. Mitochondria also generate ROS through electron-transfer intermediates intimately involved in cell death signalling pathways [4,48]. In the process of OA-induced PCD, the disruption of respiratory electron transport chains and loss of nuclear membrane integrity were detected by specific fluorescent indicators Rd123 and PI. Mitochondria undergo significant changes before classic characteristics of the nuclear membrane appear. The main cellular organelle structure was damaged, and related possible signalling pathways induced by high-dose OA caused the entire PDC. We confirmed the OA toxicity to the cells.

It was observed that cytosolic pH regulation of anion channels plays a specific role in the cytosolic pH regulation in plant cells by providing an anion shunt conductance [48]. Anion channels/transporters seem to act as key players in signalling pathways leading to the adaptation of plant cells to abiotic and biotic stresses in control of metabolism and the maintenance of electrochemical gradients. Previous research suggested that increase of anion current was a required upstream event in the signalling pathway leading to oxalate-induced cell death [16]. (2’,7’-Bis-(2-carboxyethyl)-5-(and-6)-carboxyfluorescein-acetoxymethyl (BCECF-AM) was employed to detect the possible cytoplasmic pH (pH_cyt_) oscillations in the exposure of the suspension-cultured cell to OA. The decrease in time and dose-dependent florescent density decrease indicated the intracellular acidification in the process. It mimicked pH_cy_ drops using the ammonium chloride (NH_4_Cl) method leading to more cell death than the usual condition, suggesting that the cytosolic acidification induced by OA may be critical for initiation PCD. The pretreatment of FC (as an activator of H^+^-ATPase, cytosolic alkalinization) could efficiently inhibit the cell death induced by OA with a drop of the cell death percentage by 70%; the evidence could further confirm the pH_cyt_ drop role played in the OA-induced cell death. In mammals, many experiments have demonstrated that cytoplasmic acidification is a feature of apoptosis. Several agents leading to cytoplasmic alkalinization through activation of ion channels and pumps could prevent apoptosis stimulated by intra- or extracellular elements [49].

Interestingly, some similar results were obtained in the field of plant science. The hypothesis predicts that biotic and abiotic stresses-induced cytoplasmic acidification triggers the synthesis of phytoalexins and other secondary metabolites. Cytoplasmic acidification, which caused DNA breakdown, active caspase-like enzymes, and ROS, might act as messages involved in triggering defense responses and related PCD [50].

Intracellular acidification, combined with K^+^ and Ca^2+^ flux, was regarded as an early marker of an elicitation process leading to PCD. Several studies suggested that changes in pH_cyt_ resulting from ion fluxes and H^+^-ATPase play a role [51]. Recently, members of 2ligand-gated ion channel families, glutamate receptor-like channels (GLRs) and cyclic nucleotide-gated channels (CNGCs) were implicated in immune responses. Nevertheless, more precise data are necessary to understand their direct involvement in creating Ca^2+^ signals during immune responses [52]. These results supported the view that the ion fluxes are related to the early signalling for PCD. Inward and outward rectifying K^+^ channels carrying K^+^ ions across the membrane played a critical role in regulating biochemical balance. The triggering of the HR in tobacco cells by specific bacterial pathogens required the activation of a plasma membrane K^+^/H^+^ exchanger, which needs H^+^-ATPase function [53]. Combined with these results, it might suggest that the addition of OA functioned on the K^+^/H^+^ exchanger, which decreased the K^+^ influx (Figure 7) and showed OA played a prominent inhibition role on the inward K^+^ current paralleled by the drop of efflux of H^+^. The accumulated H^+^ in the cytoplasm may contribute to the pH_cyt_ reduction. In the process, OA may also influence the phosphorylation of the H^+^-ATPase, which not only altered the activity of the K^+^/H^+^ exchanger but mediated the H^+^ extrusion from the vacuolar proton pool. The results of FC as the activator of the H^+^-ATPase, which is bound to the 14-3-3 family regulatory protein associated with the phosphorylation-dependent C-terminal end, played a role in OA-induced cytoplasmic acidification, and related PCD could further support our hypothesis [54]. Elicitor-induced Ca^2+^ spiking was one of the earliest events that acted as a master messenger for most downstream response reactions. Some studies reported that [Ca^2^^+^]_cyt_ elevation down-regulates inward-rectifying K^+^ channels and proton pumps in the plasma membrane of guard cells [55,56]. Results also showed that OA could activate the inward Ca^2+^ channel effectively by mediating the activity of H^+^-ATPase. The induced [Ca^2^^+^]_cyt_ elevation may originate from the extracellular Ca^2+^ influx or efflux of some organelles such as endoplasmic reticulum, mitochondria, which could regulate [Ca^2^^+^]_cyt_ through Ca^2+^-ATPase and Ca^2+^/H^+^ exchanger [57]. EGTA (a Ca^2+^ chelator with at least 10^4^ fold greater affinity than OA) and La^3+^ (a Ca^2+^ channel blocker) were used to pre-incubate the OA-treated cells, which could remove the possible role of extracellular Ca^2+^. The slight effect on the decrease of pH_cyt_ suggested that extracellular Ca^2+^ might not be the primary mechanism participating in the regulation of pH_cyt,_ and the chelation of OA might not be the primary function in the process of PCD. The Ca^2+^ store deletion is possible as the primary source of the [Ca^2^^+^]_cyt_ elevation. More related genic and proteinic studies are needed to illustrate these points and the oxalic acid-induced cell signal significance.

## 4. Materials and Methods

### 4.1. Chemical Materials

Cellulase RS was provided by Yakult Honsha (Tokyo, Japan). Pectolyase Y-23 was provided by Seishin Pharmaceutical (Tokyo, Japan). Fluorescein diacetate (FDA), propidium iodide (PI) and BCECF-AM were acquired from Abcam Co. Ltd. (Cambridge, MA, USA). Ascorbic acid (Vc), Mes, Hepes, Mg-ATP, BSA, EGTA, K-glutamate, Rhodamine 123 (Rd123), FC were acquired from Sigma-Aldrich (St. Louis, MI, USA). Other chemicals of analytic grade were sourced from Chinese companies. Chemicals used in this study were dissolved in water or dimethyl sulphoxide (DMSO).

### 4.2. Plant Materials

Under aseptic conditions, selected loose, light yellow or milky *P. tunicoides* callus established MS liquid culture system according to callus quality and culture medium volume 1:10 (g/mL). Cultures were incubated at 25 ± 1 °C in the dark on a rotary shaker at 110 ± 5 rpm overnight, and the suspension cells of *P. tunicoides* were harvested.

The suspension cells of *P. tunicoides* were cultured in 250 mL Erlenmeyer flasks containing 100 mL of Murashige and Skoog [58] salt solution, supplemented with 6-BA 0.5 mg/L, 2,4-D 0.5mg/L, the pH adjusted 5.8 by NaOH or HCl. Cultures were incubated at 22 ± 2 °C in the dark on a rotary shaker at 110 ± 5 rpm overnight. Cell suspensions were transferred into a new medium after 14 days using 1:5 dilutions. All experiments were performed at 22 ± 2 °C using log-phase cells (6 days after subculture).

### 4.3. Determination of Cell Viability and Death

For FDA staining, cells were incubated in an aqueous FDA solution (0.01% *w*/*v*) for 15 min at room temperature, and followed the Evans blue staining method procedure. Cells were observed under a fluorescence microscope (Motic AE31); only live cells appeared green. The cell viability was indicated as the ratio of live cells to total cells.

Evans blue staining was employed to determine cell death. The suspension cells were incubated in a 0.1% (*w*/*v*) aqueous solution of Evans blue for 10 min at room temperature. The cells were then washed with fresh MS media twice to remove unbound dye cells before observation by centrifugation at 600× *g* rpm for 5 min. Subsequently, cells were observed and counted with a bright field microscope (Motic AE31); only dead cells appeared blue. The cell death was indicated as the ratio of dead cells to total cells.

All experiments were independently replicated at least 3 times, and 500 cells were examined and analyzed statistically. Data are presented as means ± SD.

### 4.4. Rd 123 and PI Staining Procedures

Rd123 staining is plasma membrane-permeable and aggregates within mitochondria. The excited fluorescence intensity Rd123 is proportionate to the potential of electron transport chains, reflecting the functional integrity of mitochondria. We added 100 mL of Rd123 stock solution (10 mg/L) to 1 mL of cells suspension to a final concentration of 1 g/mL, and incubated at 25 °C for 15 min in a dark environment. The suspension cells were rinsed 3 times with new MS media to remove excess Rd123.

For PI staining to examine the integrity of the plasma membrane, 20 L of PI stock solution (20 g/L), which is a membrane-impermeable DNA/RNA stain, was added to 1 mL of cells suspension, and the cells were gently centrifuged and incubated in controlled darkness at 27 °C for 15 min. Subsequently, the suspension cells were rinsed 3 times with new MS media to remove excess PI. PI-negative staining cells are live cells, and PI-positive staining cells are primary cells in the late stages of PCD. All experiments were repeated 3 times.

### 4.5. Intracellular pH (pH_i_) Measurement

We dissolved 50 μg of BCECF-AM in 8.4 μL DMSO to a final concentration of 10 mmol/L and stored it at −20 °C in a controlled dark environment as stock. Further, the working solution was prepared by adding 2 μL of stock solution to 1 mL of suspension cells to a final concentration of 20 μmol/L; after incubating in the working solution for 15 min at 25 ± 1 °C, the suspension cells were rinsed 3 times in a new MS medium to remove excess BCECE-AM. These cells fluorescence emission images were acquired using a cooled charged-coupled device (CCD) camera on an inverted microscope (Motic AE31). The fluorescence intensity was calculated using software IPP 6.0 through the analysis of fluorescence images. The CCD camera was also used for bright-field images collection.

All experiments were repeated 3 times with different samples, and representative images were presented.

### 4.6. Patch Clamp and Data Acquisition

Protoplasts of *P. tunicoides* were isolated as described [59]. In the whole-cell voltage-clamp, the K^+^ and Ca^2+^ currents of *P. tunicoides* cells were recorded with an EPC-9 amplifier (Heka Instrument) described [60]. Pipettes were pulled with a vertical puller (Narishige) modified for two-stage pulls. Data were analyzed using PULSEFIT 8.7, IGOR 3.0, and ORIGIN 7.0 software. The standard solution for potassium current measurements contained 10 mM K-glutamate, 1 mM CaCl_2_, 2 mM MgCl_2_, 10 mM Mes, pH 5.5, in the bath and 80 mM K-glutamate, 1.1 mM EGTA, 5 mM Mg-ATP, 20 mM KCl, 10 mM Hepes, pH 7.2, in the pipette. The standard solution for calcium current measurements contained 100 mM CaCl_2_, 0.1 mM DTT, and 10 mM MES-Tris, pH 5.6, in the bath and 10 mM BaCl_2_, 0.1 mM DTT, 4 mM EGTA, and 10 mM HEPES-Tris, pH 7.1, in the pipette. D-sorbitol was used to adjust the osmolality of pipette and bath solutions to 400 and 500 mmol/kg, respectively.

## 5. Conclusions

The regulation and execution processes of PCD, particularly the processes induced by abiotic factors, remain unknown. PCD is a crucial process in plant development, senescence or immunity and plays an important role in the plant stress response. The processes and biochemical and molecular pathways of plant PCD induced by abiotic stress are very important for understanding the tolerance/resistance of plants to abiotic stress, enabling plant tolerance to be increased in the future by manipulating the inhibition of PCD. In a global environment with climate changes, susceptible and tolerant genotypes/species are highly desirable. Although some of the biochemical, molecular, and morphological mechanisms are known, in this paper, we focused on the PCD process, mechanisms, and induced by OA in *P. tunicoides*, which is a unique perennial medicinal plant species, including the cytoplasmic pH oscillations that are essential in the process independent of the extracellular Ca^2+^. It is conceivable that OA is correlated to ion channels located at the plasma membrane in triggering responses of plant cells to various environmental perturbations. Our study suggests that OA may influence inward K^+^ channels and Ca^2+^ channels by mediating the activation of H^+^-ATPase.

## Figures and Tables

**Figure 1 molecules-26-06957-f001:**
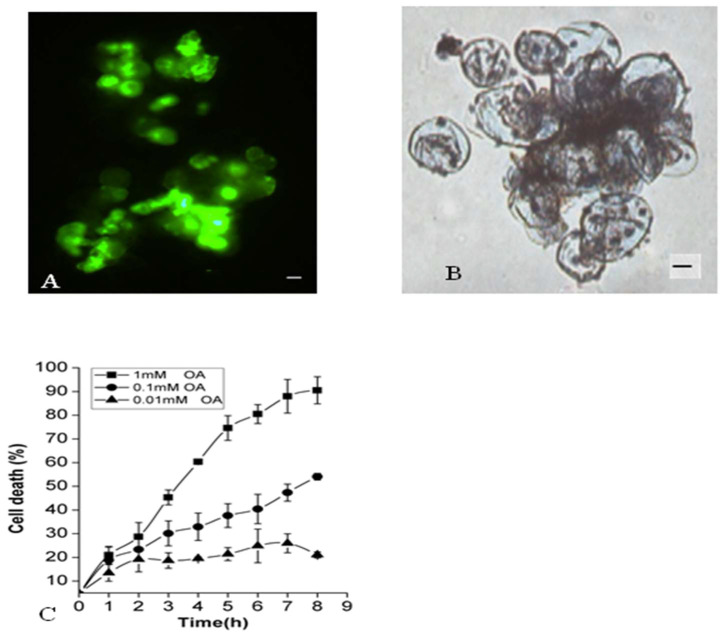
Effects of oxalic acid on *P. tunicoides* suspension cell viability. Cells were treated with 0.1 mM OA for 8 h, stained with FDA (**A**). and with 1 mM OA for 8 h, stained with Evans blue (**B**), Effect of different concentrations of OA on cell viability from 0 to 8 treatment (**C**). Bar = 10 μm. Values are expressed as means ± SEM.

**Figure 2 molecules-26-06957-f002:**
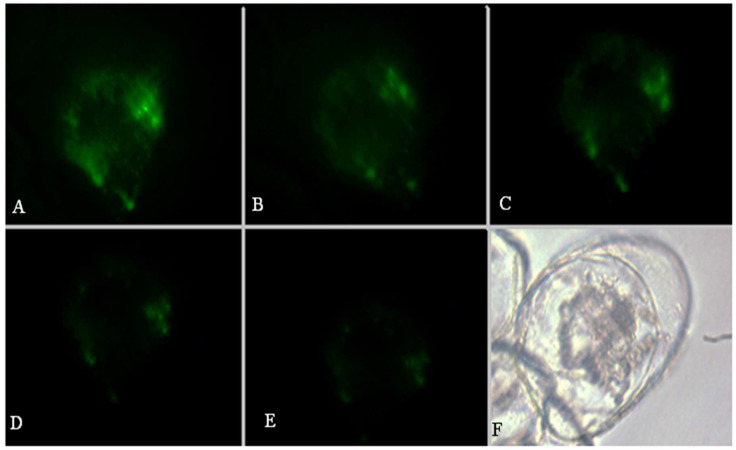
A series of images of cells were undergoing disruption of respiratory electron transport chains after treating 1 mmol/L of OA. Rd123 stained the cells. The green fluorescence of Rd123 in the mitochondria was evident (**A**). After the treatment of 1 mM OA, the fluorescence intensity began to decrease after 15 min (**B**). The boundary became obscure, and the fluorescence intensity continued to drop after 30 min (**C**). After 60 min the fluorescence intensity in the mitochondria decreased markedly (**D**). The fluorescence intensity disappeared almost entirely after 75 min (**E**). The bright-field photo of (**E**) picture (**F**).

**Figure 3 molecules-26-06957-f003:**
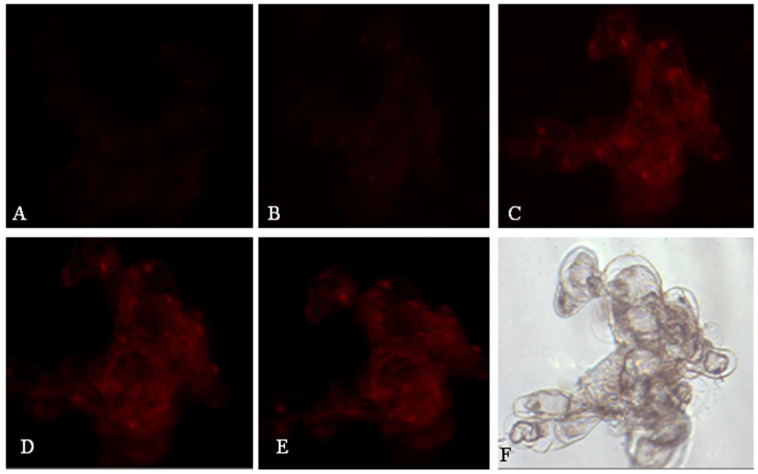
Serial fluorescence images loss of nuclear membrane integrity after the treatment of 1 mmol/L OA stained by PI. The nuclear membrane maintained integration, and weak red fluorescence could be observed (**A**). After the treatment of 1 mM OA, the fluorescence began to appear (after 1 h) (**B**). The fluorescence intensity increased because the nuclear membrane integrity was lost (after 3 h) (**C**). The fluorescence intensity in cytoplasm, especially the location of nucleus, continued to increase (after 5 h) (**D**). A large amount of PI stained the nucleus, and the fluorescence intensity was evident (after 7 h) (**E**). Microscopic was photographed under the white light field from E cells, and the cells were relatively intact (**F**).

**Figure 4 molecules-26-06957-f004:**
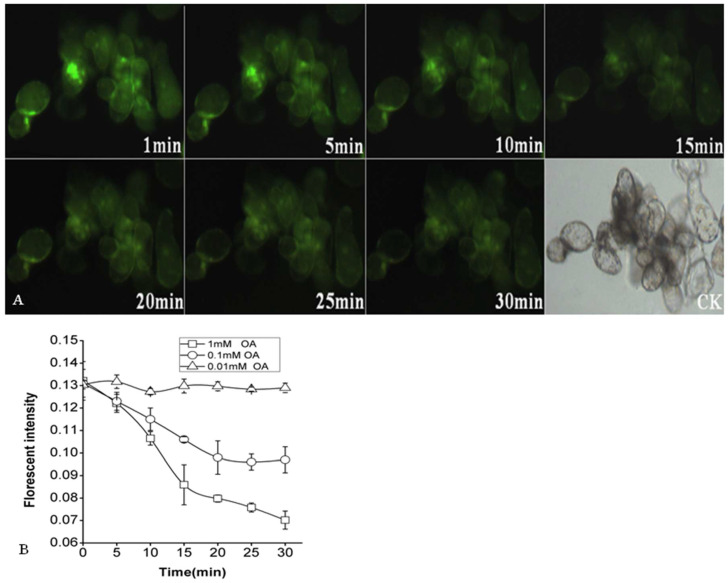
Effect of OA on *P. tunicoides* suspension cell pH_cyt_. Series of images of acidification in the cytoplasm after treating 1 mmol/L oxalic acid marked by BCECF-AM (**A**). Effect of different concentrations of OA on the pH_cyt_ (**B**). Values are expressed as means ± SEM.

**Figure 5 molecules-26-06957-f005:**
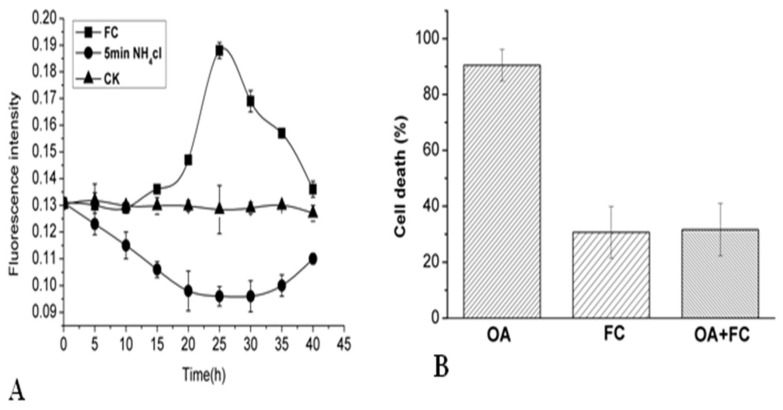
Effects of different agents on pH_cyt_ and related changes in cell viability. Effects of FC and NH_4_Cl on the pH_cyt_ (**A**). Effects of FC on the 1 mM OA-treated cell viability (**B**). Values are expressed as means ± SEM; *p* < 0.05.

**Figure 6 molecules-26-06957-f006:**
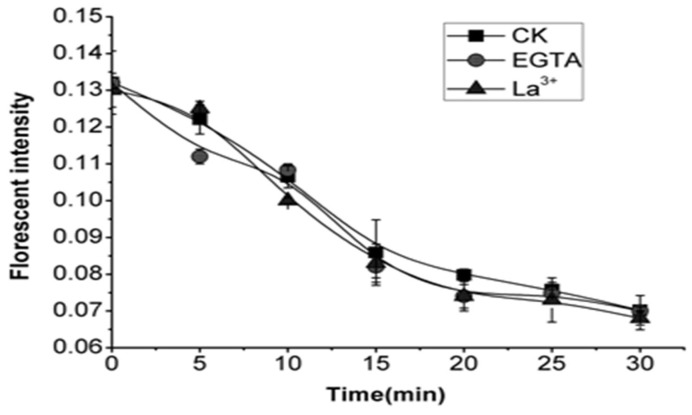
Effects of EGTA and La^3+^ on 1 mM OA-treated cell pH_cyt_.

**Figure 7 molecules-26-06957-f007:**
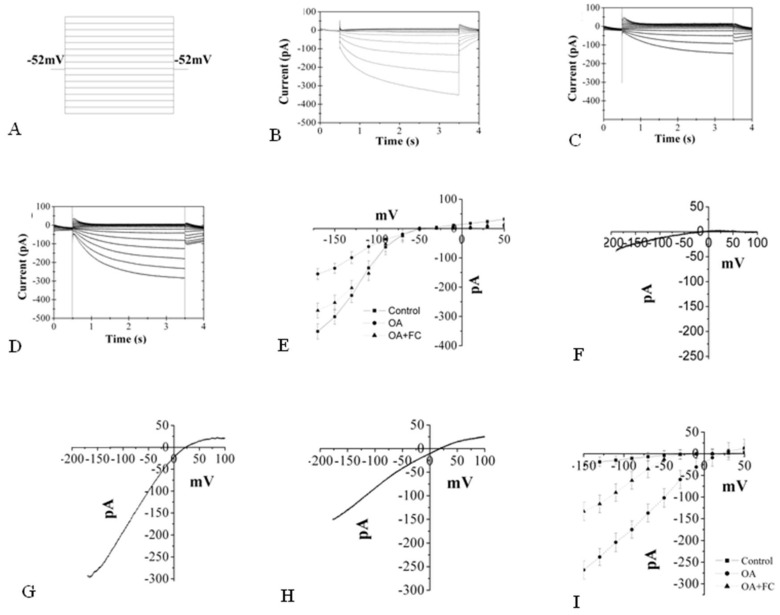
OA-induced changes in K^+^ and Ca^2+^ inward currents. The value of the resting membrane potential (Vm) of control cells in their culture medium was −52 mV (**A**). The inward K^+^ current is in normal condition (**B**). The inward K^+^ current decreased obviously (from 350 pA to 150 pA) after 1 mM OA (**C**) treatment. The inward K^+^ current recovered to 280 pA after adding FC on 1 mM OA-treated cell (**D**). Corresponding K^+^ current-voltage relationships (**E**). The inward Ca^2+^ current is in normal condition (**F**). The inward Ca^2+^ current increased to 300 pA after the treatment of 1 mM OA (**G**). The inward Ca^2+^ current dropped to 150 pA after the addition of FC on 1 mM OA-treated cell (**H**). Corresponding Ca^2+^ current-voltage relationships (**I**). Values are expressed as percentages of the control (means ± SEM).

## Data Availability

The data presented in this study are available on request from the corresponding authors.

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
