# Peer review of "Towards Understanding the Involvement of H+-ATPase in Programmed Cell Death of Psammosilene tunicoides after Oxalic Acid Application"

_molecules, 2021, doi:10.3390/molecules26226957_

Round 1

Reviewer 1 Report

The manuscript aims to show that H+-ATPase activity can be a key factor for oxalate-inducing programmed cell death (PCD) of P. tunicoides suspension cells. PCD is always an interesting subject, and studies evaluating oxalic acid-induced cell death have importance. However, I recommend improvements associated to the general presentation of the manuscript, as indicated below:

1) In the abstract, the authors affirm that "The OA treatment independently resulted in a 90%, and the FC led to 25% cell death rates, respectively. Consistently, the combined treatments caused 25% cell death rate". However, when observing the results presented in Figure 5B (and in the Results section), the correct value is not 25%. Please, revise the abstract;

2) In the legend for Figure 2, reference must be made to subFigure (F);

3) Various subsections are not numbered correctly (sub section 3.5 should be 2.5, etc). Please, revise;

4) Lines 265-267: The authors state that "The pretreatment of FC (as an activator of H+-ATPase, cytosolic alkalinization) could efficiently inhibit the cell death induced by OA with a drop of the cell death percentage by 50%". Actually, the drop of the cell death presented was larger than 50%. Please, revise it;

5) Lines 294-295: Please, a reference must be included for "... H+-ATPase, which is bound to the 14-3-3 family regulatory protein associated with the phosphorylation-dependent C-terminal end (maybe, Würtele M, Jelich-Ottmann C, Wittinghofer A, Oecking C. Structural view of a fungal toxin acting on a 14-3-3 regulatory complex. EMBO J. 2003 Mar 3;22(5):987-94. doi: 10.1093/emboj/cdg104. PMID: 12606564; PMCID: PMC150337)";

6) References 30 and 31 are out of order. Please revise the order of references;

7) An additional editing/revision of English language must be carried out.

Author Response

To: Molecules

Date: November 8, 2021

RE: Response to reviewer #1 comments

Dear Editor of Molecules

Thank you for the recommendations from the reviewers to our manuscript entitled:

" Towards Understanding the Involvement of H+-ATPase in Programmed Cell Death of Psammosilene tunicoides After Oxalic Acid Application " (Manuscript ID: molecules-1421943).

We appreciate all the comments from the three reviewers. We found the comments very helpful for improving the manuscript, and we have carefully revised the manuscript according to the suggestions. Please find our responses to the reviewers’ comments below, and all the revised parts marked in red in the revised manuscript.

Please inform us if there are any further requirements or comments.

Sincerely,

Yuan Qin

Professor,

Dean College of Life Science

Center for Genomic and Biotechnology

Haixia Institute of Science and Technology, Fujian Agriculture and Forestry University,

 Fujian, Fuzhou, China 350005

Email: yuanqin@fafu.edu.cn;

Phone: 86-591-83595382

Reviewer #1

The manuscript aims to show that H+-ATPase activity can be a key factor for oxalate-inducing programmed cell death (PCD) of P. tunicoides suspension cells. PCD is always an interesting subject, and studies evaluating oxalic acid-induced cell death have importance. However, I recommend improvements associated to the general presentation of the manuscript, as indicated below:

Response: Thank you very much for your comments and suggestion, we have already invited English native speakers to help us to improve this manuscript. We have revised ntire manuscript and the revised are marked in red.

  • In the abstract, the authors affirm that "The OA treatment independently resulted in a 90%, and the FC led to 25% cell death rates, respectively. Consistently, the combined treatments caused 25% cell death rate". However, when observing the results presented in Figure 5B (and in the Results section), the correct value is not 25%. Please, revise the abstract.

Response: We have inserted the correct % please see Line; 35

  • In the legend for Figure 2, reference must be made to sub Figure (F);

Response: We are sorry to be careless. The reference added.

  • Various subsections are not numbered correctly (sub section 3.5 should be 2.5, etc). Please, revise;

Response: the subsection has correct entire manuscript.

  • Lines 265-267: The authors state that "The pretreatment of FC (as an activator of H+-ATPase, cytosolic alkalinization) could efficiently inhibit the cell death induced by OA with a drop of the cell death percentage by 50%". Actually, the drop of the cell death presented was larger than 50%. Please, revise it;

Response: we add the correct number.

  • Lines 294-295: Please, a reference must be included for "... H+-ATPase, which is bound to the 14-3-3 family regulatory protein associated with the phosphorylation-dependent C-terminal end (maybe, Würtele M, Jelich-Ottmann C, Wittinghofer A, Oecking C. Structural view of a fungal toxin acting on a 14-3-3 regulatory complex. EMBO J. 2003 Mar 3;22(5):987-94. doi: 10.1093/emboj/cdg104. PMID: 12606564; PMCID: PMC150337)";

Response: We have done it and checked the lines in the paper thoroughly according to your suggestion.

  • References 30 and 31 are out of order. Please revise the order of references;

Response: The reference formatted with correct reference.

  • An additional editing/revision of the English language must be carried out.

Response: The manuscript was revised entirely; please see the new draft.

Reviewer 2 Report

Dear authors
It is an interesting article, but before being published in the journal Molecules, but it should be improved.

First of all, the abstract should be shortened, it is too long and cumbersome.
The introduction should be complemented by demarcating the objectives of the work.
The results should be accompanied by the respective figures and tables.
The methodology in line 366 because it appears a point 2.6 to which it refers, it does not talk about the statistical study carried out.
The conlusion is not there
The references are out of format

The abbreviations should be revised and ml should be replaced by mL, among other formatting errors.

Author Response

To: Molecules

Date: November 8, 2021

RE: Response to reviewer #2 comments

Dear Editor of Molecules

Thank you for the recommendations from the reviewers to our manuscript entitled:

" Towards Understanding the Involvement of H+-ATPase in Programmed Cell Death of Psammosilene tunicoides After Oxalic Acid Application " (Manuscript ID: molecules-1421943).

We appreciate all the comments from the three reviewers. We found the comments very helpful for improving the manuscript, and we have carefully revised the manuscript according to the suggestions. Please find our responses to the reviewers’ comments below, and all the revised parts marked in red in the revised manuscript.

Please inform us if there are any further requirements or comments.

Sincerely,

Yuan Qin

Professor,

Dean College of Life Science

Center for Genomic and Biotechnology

Haixia Institute of Science and Technology, Fujian Agriculture and Forestry University,

 Fujian, Fuzhou, China 350005

Email: yuanqin@fafu.edu.cn;

Phone: 86-591-83595382

Dear authors
It is an interesting article, but before being published in the journal Molecules, but it should be improved. First of all, the abstract should be shortened, it is too long and cumbersome.

Response: Thank you very much for your suggestion, we have revised and improved this manuscript. All the revised parts are marked

in red. The abstract was shortened to 270 words.
The introduction should be complemented by demarcating the objectives of the work

Response:
The results should be accompanied by the respective figures and tables.

Response: the tables and figures with their legends move to the main text.
The methodology in line 366 because it appears a point 2.6 to which it refers, does not talk about the statistical study carried out.

Response:
The conclusion is not there

Response: We have added the conclusion part please see Line; 454-470.
The references are out of format

Response; The reference has cited machinery. We used EndNote X9 for citation, by the way, double-checked through the manuscript. You can revise sections are marked in red.

The abbreviations should be revised and ml should be replaced by mL, among other formatting errors.

Response: we have reived it.

Reviewer 3 Report

The present manuscript aimed to evaluate the H+-ATPase activity as a key factor for oxalate-inducing programmed cell death (PCD) of Psammosilene tunicoides suspension cells. P. tunicoides is a unique perennial medicinal plant species native to the South-western regions of China. Specifically, the focus is on oxalic acid (OA) that was reported to be as an effective abiotic elicitor that enhances plants cell resistance to environmental stress. It was shown that OA significantly decreased the mitochondria activity and damaged its electron transport chain.

The work is interesting and has been carried out with adequate means. However, more related genic and proteinic studies are needed to illustrate the oxalic acid-induced cell signal significance.

  • In Section 4.2 no description of the plant is provided. Voucher specimen?
  • English can be slightly improved.

Author Response

To: Molecules

Date: November 8, 2021

RE: Response to reviewer #3 comments

Dear Editor of Molecules

Thank you for the recommendations from the reviewers to our manuscript entitled:

" Towards Understanding the Involvement of H+-ATPase in Programmed Cell Death of Psammosilene tunicoides After Oxalic Acid Application " (Manuscript ID: molecules-1421943).

We appreciate all the comments from the three reviewers. We found the comments very helpful for improving the manuscript, and we have carefully revised the manuscript according to the suggestions. Please find our responses to the reviewers’ comments below, and all the revised parts marked in red in the revised manuscript.

Please inform us if there are any further requirements or comments.

Sincerely,

Yuan Qin

Professor,

Dean College of Life Science

Center for Genomic and Biotechnology

Haixia Institute of Science and Technology, Fujian Agriculture and Forestry University,

 Fujian, Fuzhou, China 350005

Email: yuanqin@fafu.edu.cn;

Phone: 86-591-83595382

The present manuscript aimed to evaluate the H+-ATPase activity as a key factor for oxalate-inducing programmed cell death (PCD) of Psammosilene tunicoides suspension cells. P. tunicoides is a unique perennial medicinal plant species native to the Southwestern regions of China. Specifically, the focus is on oxalic acid (OA) which was reported to be an effective abiotic elicitor that enhances plants cell resistance to environmental stress. It was shown that OA significantly decreased the mitochondria activity and damaged its electron transport chain.

The work is interesting and has been carried out with adequate means. However, more related genic and proteinic studies are needed to illustrate the oxalic acid-induced cell signal significance.

Response; Thank you very much for your positive comments about our manuscript. We have substantially improved the concerned sections entire manuscript as seen in the revised version.

  • In Section 4.2 no description of the plant is provided. Voucher specimen?

Response; we have revised please see the new version.

  • English can be slightly improved.
  • Response: we improved the manuscript writing.

Reviewer 4 Report

In paragraph 4.2, authors should describe the method of preparation of the cell suspension.

In the manuscript, there is no description of the statistical approach eventually followed.

Please include a paragraph dedicated to the statistical analysis.

The statistical analysis (i.e. ANOVA with appropriate post hoc test-Newman-Keuls). These tests are to be particularly applied as regards the data reported in figures 1,4,5,6.

The figures and the legends should be moved in the results section

The format of reference #1 has to be corrected.

Author Response

To: Molecules

Date: November 8, 2021

RE: Response to reviewer #4 comments

Dear Editor of Molecules

Thank you for the recommendations from the reviewers to our manuscript entitled:

" Towards Understanding the Involvement of H+-ATPase in Programmed Cell Death of Psammosilene tunicoides After Oxalic Acid Application " (Manuscript ID: molecules-1421943).

We appreciate all the comments from the three reviewers. We found the comments very helpful for improving the manuscript, and we have carefully revised the manuscript according to the suggestions. Please find our responses to the reviewers’ comments below, and all the revised parts marked in red in the revised manuscript.

Please inform us if there are any further requirements or comments.

Sincerely,

Yuan Qin

Professor,

Dean College of Life Science

Center for Genomic and Biotechnology

Haixia Institute of Science and Technology, Fujian Agriculture and Forestry University,

 Fujian, Fuzhou, China 350005

Email: yuanqin@fafu.edu.cn;

Phone: 86-591-83595382

In paragraph 4.2, authors should describe the method of preparation of the cell suspension.

Response;  We tried to give details on that methods.

In the manuscript, there is no description of the statistical approach eventually followed. Please include a paragraph dedicated to the statistical analysis. The statistical analysis (i.e. ANOVA with appropriate post hoc test-Newman-Keuls). These tests are to be particularly applied as regards the data reported in figures 1,4,5,6.

Response: We add the description of statistics for the figures. Please see the revised version.

The figures and the legends should be moved in the results section.

Response; we shift all figures with the legends moved to the main text.

The format of reference #1 has to be corrected.

Response; The reference has cited machinery. We used EndNote X9 for citation, by the way, double-checked through the manuscript. You can revise sections are marked in red.

Round 2

Reviewer 2 Report

Dear Authors

The changes have been made, however the abstract must be at most 200 words according to the rules of the journal, reduce it and it is suitable for publication.